# Angiotensin II receptor blocker or angiotensin-converting enzyme inhibitor use and COVID-19-related outcomes among US Veterans

Catherine G. Derington[1]*, Jordana B. Cohen[2,3], April F. Mohanty[4,5], Tom H. Greene[1], James Cook[4,5], Jian Ying[1,4], Guo Wei[4], Jennifer S. Herrick[1,4], Vanessa W. Stevens[1,4], Barbara E. Jones[4,5], Libo Wang[6], Alexander R. Zheutlin[5], Andrew M. South[7,8], Thomas C. Hanff[9], Steven M. Smith[10], Rhonda M. Cooper-DeHoff[10,11], Jordan B. King[1,12], G. Caleb Alexander[13], Dan R. Berlowitz[14,15], Faraz S. Ahmad[16], M. Jason Penrod[5], Rachel Hess[1,5], Molly B. Conroy[1,5], James C. Fang[6], Michael A. Rubin[4,5], Srinivasan Beddhu[5], Alfred K. Cheung[5], Weiming Xian[15,17], William S. Weintraub[18], Adam P. Bress[1,4]

1 Department of Population Health Sciences, Division of Health System Innovation and Research, University of Utah School of Medicine, Salt Lake City, UT, United States of America, 2 Department of Medicine, Renal-Electrolyte and Hypertension Division, Perelman School of Medicine at the University of Pennsylvania, Philadelphia, PA, United States of America, 3 Department of Biostatistics, Epidemiology, and Informatics, Perelman School of Medicine, University of Pennsylvania, Philadelphia, PA, United States of America, 4 George E. Wahlen Department of Veterans Affairs Medical Center, Salt Lake City, UT, United States of America, 5 Department of Internal Medicine, University of Utah School of Medicine, Salt Lake City, UT, United States of America, 6 Department of Medicine, Division of Cardiology, University of Utah School of Medicine, Salt Lake City, UT, United States of America, 7 Department of Pediatrics, Section of Nephrology, Brenner Children's Hospital, Wake Forest School of Medicine, Winston Salem, NC, United States of America, 8 Division of Public Health Sciences, Department of Epidemiology and Prevention, Wake Forest School of Medicine, Winston Salem, NC, United States of America, 9 Department of Medicine, Division of Cardiology, Perelman School of Medicine, University of Pennsylvania, Philadelphia, PA, United States of America, 10 Department of Pharmacotherapy and Translational Research, University of Florida College of Pharmacy, Gainesville, FL, United States of America, 11 Department of Medicine, University of Florida, College of Medicine, Gainesville, FL, United States of America, 12 Institute for Health Research, Kaiser Permanente Colorado, Aurora, CO, United States of America, 13 Department of Epidemiology, Johns Hopkins Bloomberg School of Public Health, Baltimore, MD, 14 Department of Public Health; University of Massachusetts Lowell, Lowell, MA, United States of America, 15 Edith Nourse Rogers Memorial Veterans Hospital, Bedford, MA, United States of America, 16 Department of Medicine, Division of Cardiology, Northwestern University Feinberg School of Medicine, Chicago, IL, United States of America, 17 Department of Pharmacology and Experimental Therapeutics, Boston University School of Medicine, Boston, MA, United States of America, 18 MedStar Washington Hospital Center, Washington, DC, United States of America

* catherine.derington@utah.edu

**Data Availability Statement:** There are legal restrictions on sharing data publicly. The dataset used for this study contains sensitive patient

## Abstract

### Background

Angiotensin II receptor blockers (ARBs) and angiotensin-converting enzyme inhibitors (ACEIs) may positively or negatively impact outcomes in severe acute respiratory syndrome coronavirus 2 (SARS-CoV-2) infection. We investigated the association of ARB or ACEI use with coronavirus disease 2019 (COVID-19)-related outcomes in US Veterans with treated

information that is potentially identifying. The Department of Veterans Affairs (VA) restricts access to such data unless researchers meet specific criteria per VHA Directive 1605.01, Privacy and Release of Information, section 13e which includes a written request for the data, IRB approval with waiver for HIPAA authorization prior to the request for individually identifiable information, and approval by the VA. Data requests may be sent to: VA Information Resource Center (VIReC) Building 18 Hines VA Hospital (151V) 5000 S. 5th Avenue Hines, IL 60141-3030 708-202-2413 708-202- 2415 (fax) virec@va.gov.

**Funding:** This study received support from all of the following sources: National Heart, Lung and Blood Institute (NHLBI), through grant numbers R01HL139837, K01HL133468 (APB), R01HL139837 (APB), K23HL133843 (JBC), R01HL153646 (JBC), K01HL138172 (SMS), K23HL148394 (AMS), L40HL148910 (AMS), and R01HL146818 (AMS); the University of Utah Population Health Research Foundation, with funding in part from the National Center for Research Resources and the National Center for Advancing Translational Sciences, National Institutes of Health, through Grant UL1TR002538 (formerly 5UL1TR001067-05, 8UL1TR000105 and UL1RR025764); the National Institute on Aging, through grant number R01 R01AG065805 (APB); the Veterans Health Administration-Office of Health Services Research and Development, Career Development Award numbers IK2HX002609 (AFM) and IK2HX001908 (BEJ); the National Institutes of Health, through grant number 1R38HL143605-01 (ARZ); the Patient-Centered Outcomes Research Institute, through grant number CDRN-1501-26692 (RMCD); and the VA Salt Lake City Health Care System with funding from the VA Informatics and Computing Infrastructure (VINCI), through the provision of resources and facilities. CGD is a past recipient of American Heart Association grant #19POST34380226/Derington/2019.

**Competing interests:** The authors have read the journal's policy, and the authors of the study have the following competing interests to declare: APB receives research grant funding to his institution from Amarin and Amgen Inc, unrelated to the current manuscript. GCA is past Chair and current member of FDA's Peripheral and Central Nervous System Advisory Committee; has served as a paid advisor to IQVIA; is a co-founding Principal and equity holder in Monument Analytics, a health care consultancy whose clients include the life sciences industry as well as plaintiffs in opioid litigation; and is a past member of OptumRx's National P&T

hypertension using an active comparator design, appropriate covariate adjustment, and negative control analyses.

## Methods and findings

In this retrospective cohort study of Veterans with treated hypertension in the Veterans Health Administration (01/19/2020-08/28/2020), we compared users of (A) ARB/ACEI vs. non-ARB/ACEI (excluding Veterans with compelling indications to reduce confounding by indication) and (B) ARB vs. ACEI among (1) SARS-CoV-2+ outpatients and (2) COVID-19 hospitalized inpatients. The primary outcome was all-cause hospitalization or mortality (outpatients) and all-cause mortality (inpatients). We estimated hazard ratios (HR) using propensity score-weighted Cox regression. Baseline characteristics were well-balanced between exposure groups after weighting. Among outpatients, there were 5.0 and 6.0 primary outcomes per 100 person-months for ARB/ACEI (n = 2,482) vs. non-ARB/ACEI (n = 2,487) users (HR 0.85, 95% confidence interval [CI] 0.73–0.99, median follow-up 87 days). Among outpatients who were ARB (n = 4,877) vs. ACEI (n = 8,704) users, there were 13.2 and 14.8 primary outcomes per 100 person-months (HR 0.91, 95%CI 0.86–0.97, median follow-up 85 days). Among inpatients who were ARB/ACEI (n = 210) vs. non-ARB/ACEI (n = 275) users, there were 3.4 and 2.0 all-cause deaths per 100 person months (HR 1.25, 95% CI 0.30–5.13, median follow-up 30 days). Among inpatients, ARB (n = 1,164) and ACEI (n = 2,014) users had 21.0 vs. 17.7 all-cause deaths, per 100 person-months (HR 1.13, 95%CI 0.93–1.38, median follow-up 30 days).

## Conclusions

This observational analysis supports continued ARB or ACEI use for patients already using these medications before SARS-CoV-2 infection. The novel beneficial association observed among outpatients between users of ARBs vs. ACEIs on hospitalization or mortality should be confirmed with randomized trials.

## Introduction

Severe acute respiratory syndrome coronavirus 2 (SARS-CoV-2) binds to angiotensin-converting enzyme 2 (ACE2) to enter and infect host cells [1]. Under homeostasis, ACE2 converts angiotensin II to angiotensin-(1–7), reducing blood pressure and inflammation [2–6]. Angiotensin II receptor blockers (ARBs) and ACE inhibitors (ACEIs) inhibit the renin-angiotensin system and may differentially influence ACE2 expression in select organs [7, 8]. During the ongoing pandemic, there has been unprecedented interest in whether ARBs and ACEIs are beneficial or harmful in patients with SARS-CoV-2 infection or coronavirus disease 2019 (COVID-19) [9].

At least twelve randomized clinical trials are testing the effects of ARB or ACEI continuation, initiation, or discontinuation on COVID-19 outcomes, two of which are now completed and reported no increased risk of harm with continuation of ARB or ACEI therapy among hemodynamically stable adults hospitalized for COVID-19 [10, 11]. Numerous retrospective studies examined the association of ARBs and ACEIs with COVID-19 outcomes using case-control and historical cohort study designs and generally reported no excess risk of

Committee. This arrangement has been reviewed and approved by Johns Hopkins University in accordance with its conflict of interest policies. JCF serves on Data and Safety Monitoring Boards (DSMBs) for Novartis, Amgen, AstraZeneca and Boehringer-Ingelheim. The views expressed are of the authors and do not necessarily represent the views or opinions of the US Government or the US Department of Veterans Affairs. This does not alter our adherence to PLOS ONE policies on sharing data and materials. There are no patents, products in development or marketed products associated with this research to declare.

SARS-CoV-2 infection or COVID-19 severity among ARB or ACEI users compared to non-users, with some suggesting potential benefit from ARBs or ACEIs [12]. However, several of the prior observational studies contain methodological limitations that may reduce their internal validity, and thus, their clinical applicability [13, 14]. Despite the volume of published observational analyses to date (more than 80 as of Feb 15, 2020) [12], focused, high-quality observational studies are needed to complement randomized trials and guide ARB or ACEI use during the growing COVID-19 pandemic. Unfortunately, due to the short duration of the pandemic limiting sample size, it is not feasible to perform a new-user design, which is considered best practice for estimating medication effects outside of randomized trials [15]. However, unlike many prior observational analyses, we utilized several methods to increase robustness of results including an active comparator design, rigorous adjustment for a rich set of covariates, and negative control analyses to assess for residual bias.

ARBs and ACEIs are among the most commonly prescribed medications in the United States (US). The potential risks of abrupt discontinuation are enormous, and their effects on the ongoing COVID-19 pandemic are not clear. Our goal was to estimate the direction and magnitude of association between ARB and ACEI use with COVID-19-related outcomes among US Veterans with treated hypertension in the outpatient and inpatient settings using Veterans Health Administration (VHA) data and robust observational methods.

## Methods

### Data source and extraction

We used outpatient and inpatient demographic, pharmacy, clinical, and healthcare utilization data from the national VHA to conduct this retrospective cohort study. The study was pre-registered with ClinicalTrials.gov (NCT04467931), where the protocol is publicly available. The University of Utah institutional review board and Salt Lake City Veterans Affairs Health Care System Research and Development Office, approved the study with a waiver of informed consent.

### Cohort derivation

We derived separate outpatient and inpatient cohorts for analyses. Veterans were eligible for inclusion as outpatients if they had a positive SARS-CoV-2 test performed between January 19, 2020 (the day before the first confirmed COVID-19 case in the US) and October 15, 2020 but were not hospitalized in the seven days before the positive test (**S1 Fig, Panel A**); the date of the positive SARS-CoV-2 test was the index date. Veterans were eligible for inclusion as inpatients if they had a positive SARS-CoV-2 test performed between January 19, 2020 and October 15, 2020 and were hospitalized for COVID-19; the index date was the first hospital admission date on or after the date where the Veteran started receiving SARS-CoV-2 testing (**S1 Fig, Panel B**). We identified COVID-19 cases using the definitions developed by the VHA for public reporting [16, 17]. Nursing home residents could be included in the outpatient and inpatient cohorts if meeting all other eligibility criteria.

We excluded Veterans who did not meet one-year continuous enrollment criteria prior to the index date or who had data inconsistencies (e.g., not alive on index date or multiple death dates). Finally, we restricted the analyses to Veterans with treated hypertension, defined as ≥1 inpatient or ≥2 outpatient encounters with an *International Classification of Diseases*, *Ninth Revision* (ICD-9) or *Tenth Revision* (ICD-10) code (401.x, 403.0x, 403.1x, 403.9x, I10, I12.0, or I12.9) [18] using all available data pre-index, and ≥1 outpatient pharmacy fill for an antihypertensive medication in the 104-day pre-index period (**S1 Table**). We selected 104 days to

account for patients who receive 90-day supply prescriptions, plus a 14-day grace period to account for modest non-adherence.

## Exposures

For both outpatient and inpatient cohorts, we assigned Veterans to exposure groups based on ≥1 outpatient pharmacy antihypertensive medication fill in the 104-day pre-index period. We compared two medication exposures among outpatients and inpatients, separately. First, we compared current users of any ARB and/or ACEI, irrespective of concurrent non-ARB/ACEI antihypertensive therapy, to Veterans using neither an ARB nor ACEI (current user of an ARB/ACEI-based regimen vs. non-ARB/ACEI-based regimen, **S2 Table**). For this comparison, because diagnoses other than hypertension (i.e., compelling indications) for ARB or ACEI use would introduce confounding by indication, we excluded Veterans with compelling indications defined in the 2017 American College of Cardiology/American Heart Association blood pressure guidelines [19] (i.e., diabetes, chronic kidney disease, coronary heart disease, heart failure with reduced ejection fraction, or prior stroke). We choose, a priori, to restrict the study population for this comparison to only those without compelling indications because we believe that the confounding by indication for this specific comparison (ACEI/ARB users compared to non-users) would be insurmountable.

Our second exposure compared current users of an ARB vs. ACEI (current user of an ARB vs. an ACEI). For this comparison, we included Veterans with and without compelling indications for ARBs and ACEIs, and we excluded Veterans who were current users of both an ARB and ACEI. For this comparison, restriction to those without compelling indications is not needed because these compelling indications are shared between both medications; ACEIs and ARBs are used interchangeably as they are thought to be equivalent in benefit and safety [19, 20]. As such, for this comparison, we included Veterans with and without compelling indications for ARBs and ACEIs, and we excluded Veterans who were current users of both an ARB and ACEI. Due to the small sample size of inpatient Veterans without compelling indications who were ARB/ACEI- vs. non-ARB/ACEI-based antihypertensive regimen users, we considered comparisons of these two groups to be exploratory.

## Outcomes

For outpatients, the primary outcome was first occurrence of all-cause hospitalization or all-cause mortality, and secondary outcomes were all-cause hospitalization, all-cause mortality, and ICU admission, separately (definitions in **S1 Table**). For outpatients, Veterans were censored at the first occurrence of an outcome or October 15, 2020. For inpatients, the primary outcome was first occurrence of all-cause mortality, and secondary outcomes were ICU admission, dialysis, or mechanical ventilation, separately. For inpatients, Veterans were censored at the first occurrence of an outcome, 30 days' follow-up, or October 15, 2020.

## Covariates

We determined baseline demographics, vital signs, laboratory measurements, and concomitant medications within the one year prior to the index date (i.e., the "pre-index period"; **S1 Table**). We defined vital signs and laboratory results using the value closest to the index date in the pre-index period, and we defined comorbid conditions using all available claims and medical record data prior to the index date (**S1 Table**). We also calculated a Charlson Comorbidity Index [21].

## Propensity score estimation

To adjust for measured confounding among outpatients and inpatients separately, we generated propensity scores (PS) to estimate the probability of being an ARB/ACEI- vs. non-ARB/ACEI-based antihypertensive regimen user and, separately, of being an ARB user vs. being an ACEI user, using logistic regression with all baseline covariates included in the PS model (**S3 Table**) [22]. The regions of common support were assessed using histograms of the PS between medication exposure groups (**S2-S3 Figs**). Next, using the PS, we calculated each Veteran's matching weight [23] and verified covariate balance in the matching weighted populations using absolute standardized mean differences [24].

## Statistical analyses

All analyses were performed in the outpatient and inpatient cohorts, separately. We compared patient characteristics between current users of ARB/ACEI- vs. non-ARB/ACEI-based antihypertensive regimens and ARB users vs. ACEI users before and after weighting. To account for missing data (**S4-S7 Tables**), we used multiple imputation with chained equations to provide 10 imputed datasets using Rubin's formulae [25, 26]. We estimated hazard ratios (HR) for the associations between the medication exposure groups and outcomes using matching-weighted Cox regression. The use of matching-weights provides a weighted analogue to 1:1 paired PS matching. We applied a two-sided alpha of 0.05 for all hypothesis tests, without correction for multiple comparisons.

## Subgroup and sensitivity analyses

All analyses, including fitting logistic regression models to estimate PS, were repeated in subgroups according to age, sex, race-ethnicity, body mass index, and the number of antihypertensive medication classes being taken.

In sensitivity analyses, we repeated Cox regression analyses comparing medication regimens using several different covariate adjustment strategies (**Supplemental Methods in S1 File**). To investigate if the observed associations were sensitive to the definition of the medication exposure window, we repeated all analyses with several alternative definitions (see S1 File). Finally, we repeated analyses using a composite of gastrointestinal bleeding or urinary tract infection as negative control outcomes.

All analyses were completed using R v.4.0.2 (R Foundation for Statistical Computing, Vienna, Austria).

# Results

## Patient characteristics

After applying all inclusion and exclusion criteria, a total of 4,969 outpatient Veterans without compelling indications were included for analysis, of which there were 2,482 and 2,487 users of an ARB/ACEI- and non-ARB/ACEI-based antihypertensive regimen, respectively (**S4 Fig**). Of 13,581 outpatient Veterans with and without compelling indications, 4,877 were current users of an ARB, and 8,704 were current users of an ACEI.

Of 485 inpatient Veterans without compelling indications, there were 210 and 275 users (i.e., filled a prescription in the 104 days prior to the admission) of an ARB/ACEI- and non-ARB/ACEI-based antihypertensive regimen, respectively (**S4 Fig**). Of 3,178 inpatient Veterans with and without compelling indications, 1,164 were current users of an ARB, and 2,014 were current users of an ACEI.

**Table 1. Matching weight-adjusted incidence rates and hazard ratios for the primary and secondary outcomes among outpatient Veterans with treated hypertension who were current users of an ARB/ACEI- vs. non-ARB/ACEI-based antihypertensive regimen and ARB user vs. ACEI user, separately.**

| Outcome | ARB/ACEI- vs. non-ARB/ACEI-based antihypertensive regimen comparison (n = 4,969) | | | | ARB user vs. ACEI user comparison (n = 13,581) | | | |
|---|---|---|---|---|---|---|---|---|
| | ARB/ACEI | Non-ARB/ACEI | Hazard Ratio | | ARB | ACEI | Hazard Ratio | |
| | (n = 2,482) | (n = 2,487) | (95% CI) | p-value | (n = 4,877) | (n = 8,704) | (95% CI) | p-value |
| *Primary Outcome* | | | | | | | | |
| All-cause hospitalization or all-cause mortality | 314 (5.0) | 395 (6.0) | 0.85 (0.73–0.99) | 0.035 | 1,435 (13.2) | 2,467 (14.8) | 0.91 (0.86–0.97) | 0.002 |
| *Secondary Outcomes* | | | | | | | | |
| All-cause hospitalization | 273 (4.3) | 344 (5.1) | 0.86 (0.73–1.01) | 0.07 | 1,254 (10.9) | 2,146 (12.3) | 0.91 (0.85–0.97) | 0.005 |
| All-cause mortality | 61 (0.8) | 65 (0.9) | 0.95 (0.66–1.36) | 0.77 | 386 (2.7) | 628 (2.8) | 0.95 (0.84–1.08) | 0.44 |
| ICU admission | 61 (0.9) | 49 (0.7) | 1.29 (0.87–1.92) | 0.21 | 257 (1.8) | 491 (2.1) | 0.84 (0.71–0.98) | 0.024 |

Numbers in table are expressed as unweighted frequency of event (weighted rate per 100 person-months).

ACEI: angiotensin-converting enzyme inhibitor; ARB: angiotensin II receptor blocker; CI: confidence interval; ICU: intensive care unit

Among both outpatients and inpatients, the majority of ARB/ACEI, non-ARB/ACEI, ARB, and ACEI users were men, obese, and had comorbid depression (**S4-S7 Tables**). The majority of outpatients and inpatients who were ARB users and ACEI users also had diabetes and/or chronic kidney disease. There were no differences in baseline characteristics after weighting (**S5 Fig, Panels A-D**, all absolute standardized mean differences <0.1).

## Outcomes

Over a median 87 days of follow-up among outpatients, there were 5.0 vs. 6.0 all-cause hospitalizations or deaths per 100 person-months among ARB/ACEI-based users vs. non-ARB/ACEI-based regimen users (HR 0.85, 95% CI 0.73–0.99; **Table 1** and **Fig 1** **Panel A**). When comparing ARB users vs. ACEI users in outpatients, there were 13.2 vs. 14.8 all-cause hospitalizations or deaths per 100 person-months over a median 85 days of follow-up (HR 0.91, 95% CI 0.86–0.97; **Table 1** and **Fig 1** **Panel B**). There was a 9.0% lower relative risk of all-cause hospitalization (HR 0.91, 95% CI 0.85–0.97) and 16.0% lower relative risk of ICU admission (HR 0.84, 95% CI 0.71–0.98) among outpatient ARB users vs. ACEI users (**S6 Fig**).

Among inpatients, over a median 30 days of follow-up, there was no statistically significant difference in all-cause mortality events in the exploratory analysis of ARB/ACEI-based users (i.e., filled a prescription in the 104 days prior to the admission) vs. non-ARB/ACEI-based regimen users (HR 1.25, 95% CI 0.30–5.13; **Table 2** and **Fig 2** **Panel A**). Over a median 30 days of follow-up, there were 21.0 vs. 17.7 all-cause deaths among inpatient ARB users vs. ACEI users (HR 1.13, 95% CI 0.93–1.38; **Table 2** and **Fig 2** **Panel B**). There were no statistically significant differences between exposure groups for secondary outcomes of ICU admission, dialysis, and mechanical ventilation (**Table 2** and **S7 Fig**).

## Subgroup and sensitivity analyses

Among outpatients taking less than three antihypertensive medication classes, users of an ARB/ACEI- vs. non-ARB/ACEI-based regimen appeared have a stronger association with all-cause hospitalization or mortality (HR 0.74, 95% CI 0.63–0.88) compared to those taking three or more antihypertensive medication classes (HR 1.52, 95% CI 0.91–2.71, p-value for interaction 0.003, **S8 Table**). Outpatient non-Hispanic Black Veterans taking an ARB vs. an ACEI also appeared to have a stronger association with all-cause hospitalization or mortality (HR 0.85, 95% CI 0.77–0.93) compared to Veterans of other race-ethnicities (HR 0.94, 95% CI

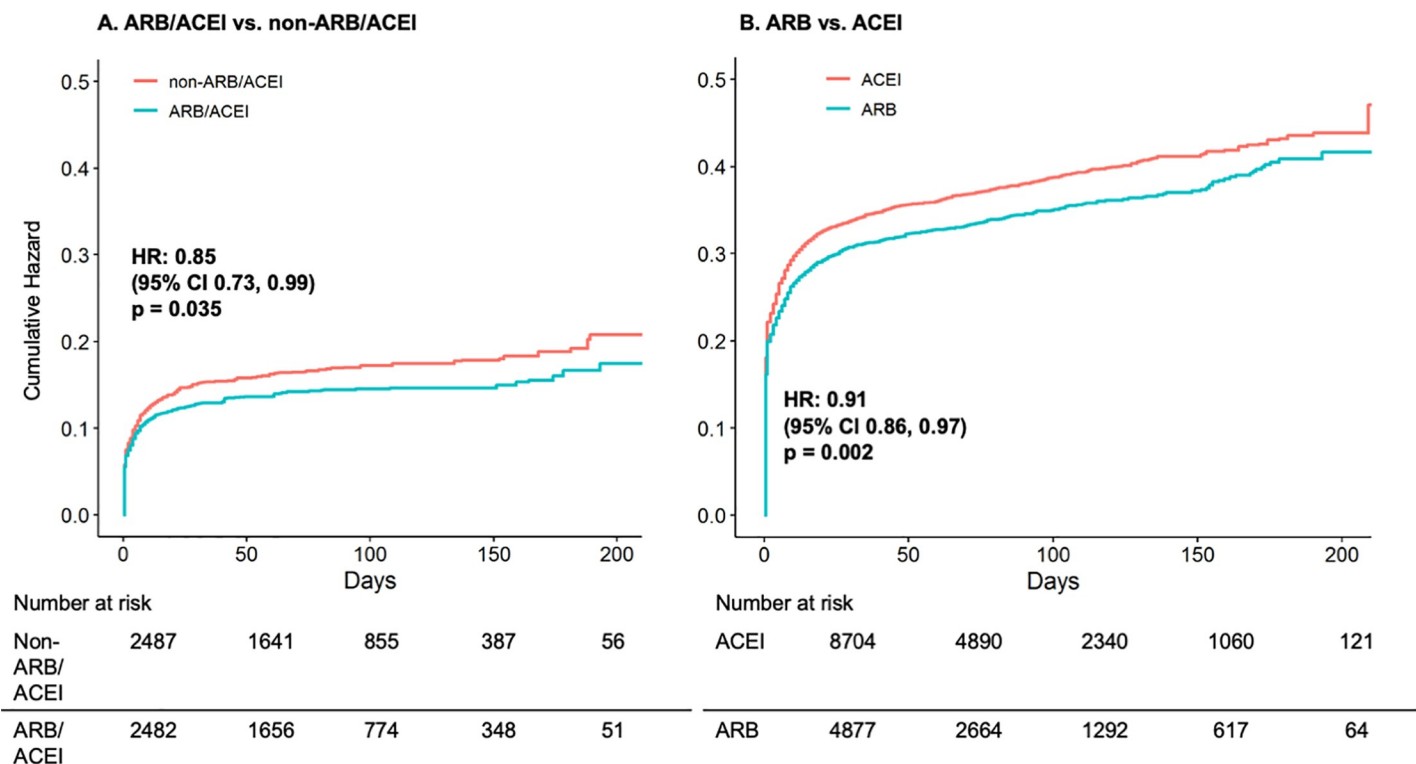

**Fig 1.** Matching weight-adjusted cumulative hazard curves for all-cause hospitalization or mortality among outpatient Veterans who were current users of an ARB/ACEI- vs. non-ARB/ACEI-based antihypertensive regimen (Panel A) and ARB user vs. ACEI user (Panel B). Cumulative incidence curves demonstrating the risk for all-cause hospitalization or mortality in outpatients between exposure groups were generated from matching-weighted Cox regression models and bootstrap resampling with 2,000 iterations. Veterans are censored at the first occurrence of all-cause hospitalization, all-cause mortality, or October 15, 2020. Numbers at risk in table are unweighted. *Abbreviations*: ACEI: angiotensin-converting enzyme inhibitor; ARB: angiotensin II receptor blocker; CI: confidence interval; HR: hazard ratio.

0.87–1.02, p-value for interaction 0.036, **S9 Table**). Conversely, inpatient non-Hispanic Black Veterans appeared to have a 48% higher relative risk of all-cause mortality if taking an ARB vs. an ACEI (HR 1.48, 95% CI 1.06–2.08), which was not observed among other race-ethnicities (HR 0.95, 95% CI 0.72–1.24, p-value for interaction 0.025, **S10 Table**). Sample size was not adequate to allow for subgroup analyses for the inpatient ARB/ACEI- vs. non-ARB/ACEI-based antihypertensive regimen comparison.

Sensitivity analysis using different covariate adjustment strategies and varying definitions of the medication exposure window did not materially change the results in either outpatients or inpatients (**S11-S15 Tables**). There was no association between any of the main exposures in either outpatients or inpatients for the negative control outcomes (**S13-S14 Tables**).

## Discussion

This historical cohort study of ARB and ACEI use among outpatient and inpatient Veterans with treated hypertension positive for SARS-CoV-2 offers at least two main findings. First, among outpatients, current use of an ARB/ACEI was associated with a 15% lower relative risk of COVID-19-related outcomes compared to a non-ARB/ACEI-based antihypertensive medication regimen. Second, this is one of the first reports demonstrating that among outpatients, ARB users had a 9% lower relative risk of all-cause hospitalization or mortality compared to ACEI users. These findings support continued use of renin-angiotensin system blockade for (1) outpatients already using an ARB or ACEI, and (2) continued use of an ARB for

**Table 2. Matching weight-adjusted incidence rates and hazard ratios for the primary and secondary outcomes among inpatient Veterans with treated hypertension who were current users of an ARB/ACEI- vs. non-ARB/ACEI-based antihypertensive regimen and ARB user vs. ACEI user, separately.**

| Outcome | ARB/ACEI- vs. non-ARB/ACEI-based antihypertensive regimen comparison* (n = 485) | | | | ARB user vs. ACEI user comparison (n = 3,178) | | | |
|---|---|---|---|---|---|---|---|---|
| | ARB/ACEI | Non-ARB/ACEI | Hazard Ratio | | ARB | ACEI | Hazard Ratio | |
| | (n = 210) | (n = 275) | (95% CI) | p-value | (n = 1,164) | (n = 2,014) | (95% CI) | p-value |
| *Primary Outcome* | | | | | | | | |
| All-cause mortality | 8 (3.4) | 6 (2.0) | 1.25 (0.30–5.13) | 0.76 | 168 (21.0) | 242 (17.7) | 1.13 (0.93–1.38) | 0.23 |
| *Secondary Outcomes* | | | | | | | | |
| ICU admission | 48 (33.0) | 40 (19.5) | 1.05 (0.54–1.94) | 0.94 | 264 (26.4) | 497 (24.5) | 0.92 (0.79–1.08) | 0.29 |
| Dialysis | 0 (0.0) | 2 (0.2) | n/a | n/a | 33 (4.6) | 448 (4.5) | 1.08 (0.84–1.40) | 0.54 |
| Mechanical ventilation | 11 (4.8) | 16 (5.9) | 0.67 (0.22–2.03) | 0.47 | 185 (24.5) | 313 (23.9) | 0.96 (0.80–1.15) | 0.65 |

Numbers in table are expressed as unweighted frequency of event (weighted rate per 100 person-months).

*Exploratory due to small sample size.

ACEI: angiotensin-converting enzyme inhibitor; ARB: angiotensin II receptor blocker; CI: confidence interval; ICU: intensive care unit

outpatients using an ARB prior to testing positive for SARS-CoV-2. For current outpatient users of ACEIs prior to testing positive for SARS-CoV-2 in clinical practice, the HR of 0.91 for all-cause hospitalization or mortality with ARB vs. ACEI is insufficiently far from 1.0 to recommend converting from an ACEI to an ARB given the prevalent user design limitations.

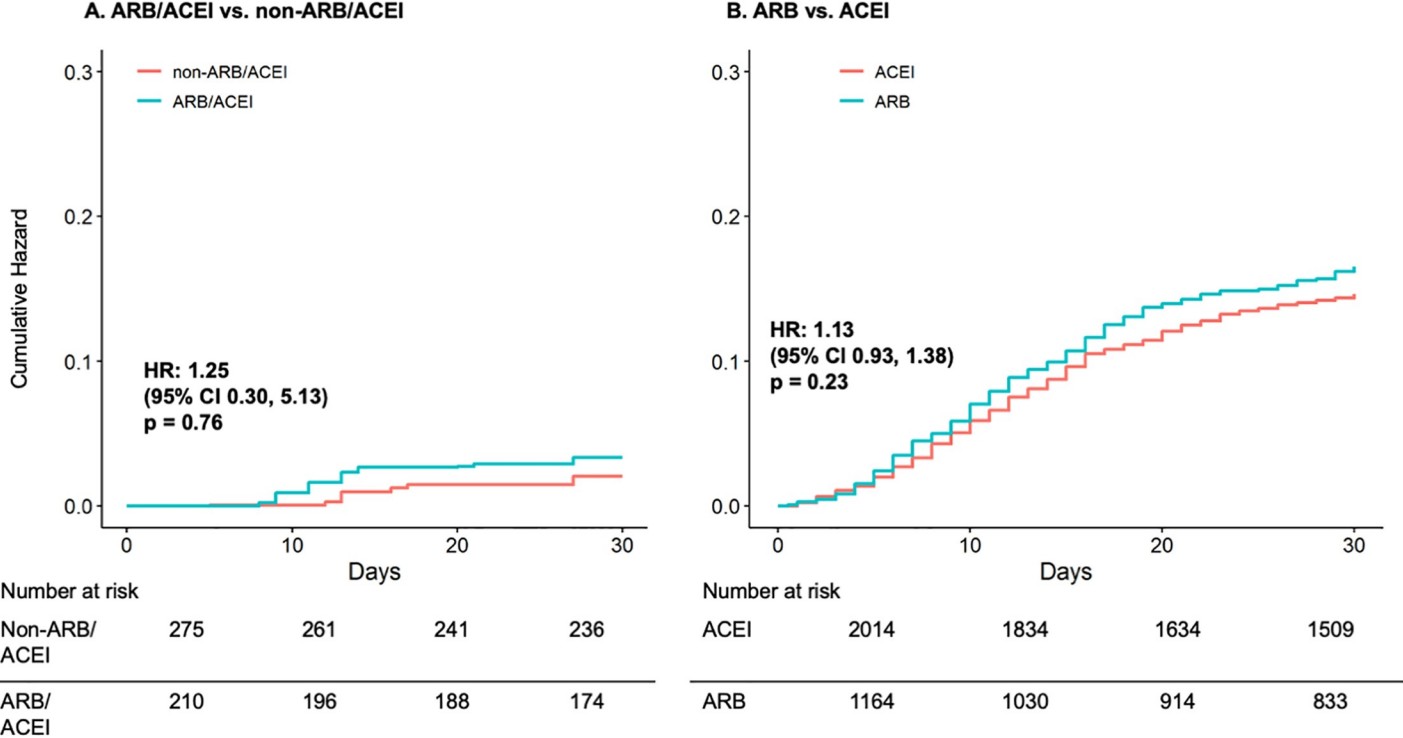

**Fig 2.** Matching weight-adjusted cumulative hazard curves for all-cause mortality among inpatient Veterans who were current users of an ARB/ACEI- vs. non-ARB/ACEI-based antihypertensive regimen (Panel A) and ARB user vs. ACEI user (Panel B). Cumulative incidence curves demonstrating the risk for all-cause mortality in inpatients between exposure groups were generated from matching-weighted Cox regression models and bootstrap resampling with 2,000 iterations. Veterans are censored at the first occurrence of all-cause mortality, 30 days' follow-up, or October 15, 2020. Numbers at risk in table are unweighted. Due to small sample size, results displayed in Panel A are exploratory. *Abbreviations*: ACEI: angiotensin-converting enzyme inhibitor; ARB: angiotensin II receptor blocker; CI: confidence interval; HR: hazard ratio.

Randomized trials are needed to confirm the observed potential beneficial effects of ARBs compared to ACEIs on COVID-19-related outcomes.

The mechanistic basis for the potential effects of ARB or ACEI use on COVID-19-related outcomes is centered around the physiology of ACE2, the protein which SARS-CoV-2 uses for host cell entry. ACE2 upregulation in the lungs by ARBs or ACEIs could enhance SARS-CoV-2 interaction with its receptor, increase infection of type II alveolar epithelial cells, and increase COVID-19 severity. The contrasting hypothesis maintains that ARBs or ACEIs could mitigate viral-induced inflammatory responses by reducing available angiotensin II or by blocking angiotensin II's actions via the angiotensin II type 1 receptor ($AT_1R$), thus shifting the balance of the renin-angiotensin system back towards ACE2 and angiotensin-(1–7). This would, thereby, promote anti-inflammatory and anti-fibrotic effects in the lungs and possibly mitigate SARS-CoV-2-induced acute lung injury [2, 27]. Animal experiments and human studies have not conclusively demonstrated the presence, direction, or magnitude of the association between ARBs or ACEIs and ACE2 levels in the circulation or tissue expression, especially in the lungs [7, 8, 28–36].

Two small, randomized trials reported no increased risk of harm with continuation of ARB or ACEI therapy among hemodynamically stable adults hospitalized for COVID-19 [10, 11]. The current report adds to these smaller trials by reinforcing the randomized trial results in a larger scale observational study in a diverse population. Also, our analysis evaluated SARS-CoV-2+ outpatients' risk of developing COVID-19, a population not tested in these trials. Furthermore, our inpatient cohort was older, and our analysis directly compared ARB vs ACEI users, which these smaller trials were not adequately powered to evaluate. High-quality observational studies have potential to augment trial results and provide helpful insights into the association of ARB and ACEI use and COVID-19 outcomes.

The current analysis may be the first observational study to show a protective association among outpatients between current users of an ARB compared to ACEI on COVID-19-related outcomes. Current users of an ARB in the outpatient cohort had a 9% lower relative risk of all-cause hospitalization or mortality and 16% lower relative risk of ICU admission compared to ACEI users. There is a mechanistic basis for the differential effects between ARBs and ACEIs. It has been hypothesized that ARBs and ACEIs have differential effects on ACE2 [7, 8]. In healthy adult humans, ACEIs, but not ARBs, may increase duodenal ACE2 expression [37]. In animals, ARB-mediated $AT_1R$ antagonism reduces peripheral and pulmonary inflammation, fibrosis, and edema [38, 39]. In addition, angiotensin II can induce beneficial effects through the $AT_2R$, and ARBs and ACEIs differentially activate secondary pathways within the renin-angiotensin system, leading to alternative angiotensin II generation and metabolism [2]. Recent data suggest that bradykinin-mediated inflammation may contribute to adverse outcomes in patients with COVID-19 [40]. Because ACEIs inhibit bradykinin metabolism, this is another plausible mechanism for differences in COVID-19-related outcomes between ACEI users and ARB users.

Understanding the extent to which use of ARBs and ACEIs is beneficial or harmful in COVID-19 is a complicated yet critical knowledge gap because hypertension is highly prevalent among Americans with more severe COVID-19-related outcomes [9]. Ideally, we would have larger randomized trials to address these gap. The recently completed trials and prior observational investigations have generally found no association between users of an ARB/ACEI vs. non-ARB/ACEI and COVID-19-related outcomes [12]. However, in contrast to the prior analyses largely conducted among inpatients or those with and without COVID-19, the current analysis identified a significant 15% reduction in all-cause hospitalization or mortality among SARS-CoV-2-positive outpatients with hypertension taking ARB/ACEI- vs. non-ARB/ACEI-based regimens. The difference in results may be due to a larger sample size in the

current analysis (i.e., increased power), more specific population (i.e., only those with treated hypertension), use of a PS matching-weighted analysis, or differences in exposure definitions compared to prior analyses. The heterogeneity in study design, study populations, exposure definitions, and statistical analyses may explain contrasting results between observational research investigating ARB and ACEI use and COVID-19-related outcomes.

In analyses evaluating ARB vs. ACEI use among inpatients, we did not find evidence of a significant association between ARB vs. ACEI users and all-cause mortality. This is in contrast to the significantly lower risk of all-cause hospitalization or mortality among ARB users compared to ACEI users in outpatients. One explanation for this inconsistency is the concept of "depletion of the susceptibles," in which ARB users who are susceptible to the beneficial effects of ARBs on all-cause hospitalization or mortality were less likely to be hospitalized, and therefore, were less likely to be included in the inpatient analysis [14]. The restriction of the inpatients to those who did not benefit from ARBs as outpatients could bias results towards observing a benefit in the ACEI group. Accordingly, we caution against over-interpretation of the directionality of the relationship between ARBs vs. ACEIs in inpatients with COVID-19. Additionally, because we ascertained ARB and ACEI use in the three months prior to SARS-CoV-2 infection, our results should not be applied to acute changes in medication therapy at the time of COVID-19 infection or during hospitalization.

The current analysis has several strengths. This was a highly focused report with two primary exposures and two cohorts utilizing detailed and validated electronic health record and pharmacy data from the largest integrated health care delivery system in the US. We included several sensitivity analyses to assess for residual confounding and robustness of results. The current study also has several limitations. The external validity of our findings may be limited by a predominantly male and white population. Because we restricted the analysis comparing ARB/ACEI- vs. non-ARB/ACEI-based regimens users to only those without a compelling indication for an ARB or an ACEI to increase internal validity, our results may not be generalizable to adults with diabetes, chronic kidney disease, coronary heart disease, heart failure with reduced ejection fraction, or prior stroke. The study question necessitated a prevalent-user design, which could introduce bias. Our active-comparator design and methods of covariate adjustment mitigate these sources of bias, as is evident through the balance in exchangeability achieved between exposure groups before and after weighting. In addition, the results of the negative control analyses suggested a low risk of substantial residual bias. We were not able to estimate the impact of initiating ARB or ACEI in patients not previously receiving treatment, stopping treatment in inpatients already receiving treatment, or, finally, switching patients between ARB and ACEI after developing COVID-19; our findings can be interpreted within the clinical question of continuing an ARB vs. continuing an ACEI during infection with SARS-CoV-2. We did not capture healthcare utilization external to the VHA, however, we do not believe there would be differential ascertainment by exposure groups. We did not adjust for multiple comparisons. Instead, consistent with the recommendations from Rothman [41], Perneger [42], Feise [43], and Althouse [44], we provided extensive information and detail on the study design, methods, and statistical analyses and report effect sizes, confidence intervals, and p-values so that the readers can make their own interpretation of our findings in the context of the other recent reports on this topic. Data on symptoms at the time of *SARS-CoV-2* testing were unavailable. Due to regional differences in the acceleration of COVID-19 cases and resource availability in the US, decisions on which patients with COVID-19 were admitted likely varied by facility and over time according to clinical judgment, local bed availability and patient acuity. Therefore, the severity of hospitalized patients with COVID-19 included in the inpatient analysis may have varied by region and over time.

## Conclusion

COVID-19 is a persistent public health crisis that has infected more than 27 million and killed more than 475,000 US adults as of February 15, 2020. More than 45 million US adults use ARBs and ACEIs annually. Unlike prior randomized trials and observational analyses, we provide novel evidence that for SARS-CoV-2-positive outpatients, use of an ARB compared to ACEI in the three months prior to infection was associated with significantly lower risk of all-cause hospitalization or mortality. Randomized trials are needed to confirm the signal of potential beneficial effects of ARBs compared to ACEIs on COVID-19-specific outcomes. We also confirmed results of prior randomized trials and observational reports which found no evidence of a harmful association of ARBs or ACEIs with adverse COVID-19-related outcomes compared to use of non-ARB/ACEI antihypertensive medications. In light of the serious risks of discontinuation of ARBs and ACEIs in patients with hypertension, our findings support professional society recommendations to continue ARB or ACEI treatment among patients with hypertension in the absence of an established indication to discontinue them.

## Supporting information

**S1 File.**
(DOCX)

## Acknowledgments

The authors would like to acknowledge the team who developed and deployed the VA's National Surveillance Tool, which informed many of the data elements for this research. This work was also supported using resources and facilities at the VA Salt Lake City Health Care System with funding from the VA Informatics and Computing Infrastructure. Finally, the support and resources from the Center for High Performance Computing at the University of Utah are gratefully acknowledged.

## Author Contributions

**Conceptualization:** Catherine G. Derington, Jordana B. Cohen, Tom H. Greene, Vanessa W. Stevens, Barbara E. Jones, Adam P. Bress.

**Data curation:** Catherine G. Derington, Tom H. Greene, James Cook, Guo Wei, Vanessa W. Stevens, Adam P. Bress.

**Formal analysis:** Tom H. Greene, Jian Ying, Guo Wei, Jennifer S. Herrick, Barbara E. Jones, M. Jason Penrod, Adam P. Bress.

**Funding acquisition:** Jennifer S. Herrick, M. Jason Penrod, Adam P. Bress.

**Investigation:** Catherine G. Derington, Jordana B. Cohen, April F. Mohanty, Tom H. Greene, James Cook, Jian Ying, Guo Wei, Jennifer S. Herrick, Barbara E. Jones, Libo Wang, Alexander R. Zheutlin, Andrew M. South, Thomas C. Hanff, Steven M. Smith, Rhonda M. Cooper-DeHoff, Jordan B. King, G. Caleb Alexander, Dan R. Berlowitz, Faraz S. Ahmad, Rachel Hess, Molly B. Conroy, James C. Fang, Michael A. Rubin, Srinivasan Beddhu, Alfred K. Cheung, Weiming Xian, William S. Weintraub, Adam P. Bress.

**Methodology:** Jordana B. Cohen, April F. Mohanty, Tom H. Greene, James Cook, Jian Ying, Barbara E. Jones, Libo Wang, Alexander R. Zheutlin, Andrew M. South, Thomas C. Hanff, Steven M. Smith, Rhonda M. Cooper-DeHoff, Jordan B. King, G. Caleb Alexander, Dan R.

Berlowitz, Faraz S. Ahmad, M. Jason Penrod, Rachel Hess, Molly B. Conroy, James C. Fang, Michael A. Rubin, Srinivasan Beddhu, Alfred K. Cheung, Weiming Xian, William S. Weintraub, Adam P. Bress.

**Project administration:** Alexander R. Zheutlin, Alfred K. Cheung, Adam P. Bress.

**Resources:** Adam P. Bress.

**Software:** Guo Wei.

**Supervision:** Catherine G. Derington, Tom H. Greene, Vanessa W. Stevens, Barbara E. Jones, Adam P. Bress.

**Validation:** Vanessa W. Stevens.

**Visualization:** Catherine G. Derington, Jordana B. Cohen, April F. Mohanty, James Cook, Libo Wang, Jordan B. King, M. Jason Penrod.

**Writing – original draft:** Catherine G. Derington, Adam P. Bress.

**Writing – review & editing:** Catherine G. Derington, Jordana B. Cohen, April F. Mohanty, Tom H. Greene, James Cook, Jian Ying, Guo Wei, Jennifer S. Herrick, Vanessa W. Stevens, Barbara E. Jones, Libo Wang, Alexander R. Zheutlin, Andrew M. South, Thomas C. Hanff, Steven M. Smith, Rhonda M. Cooper-DeHoff, Jordan B. King, G. Caleb Alexander, Dan R. Berlowitz, Faraz S. Ahmad, M. Jason Penrod, Rachel Hess, Molly B. Conroy, James C. Fang, Michael A. Rubin, Srinivasan Beddhu, Alfred K. Cheung, Weiming Xian, William S. Weintraub, Adam P. Bress.

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
