## [Decision Letter · Decision Letter 0]

10 Feb 2021

PONE-D-21-01248

Angiotensin II receptor blocker or angiotensin-converting enzyme inhibitor use and COVID-19-related outcomes among US Veterans

PLOS ONE

Dear Dr. Bress,

Thank you for submitting your manuscript to PLOS ONE. After careful consideration, we feel that it has merit but does not fully meet PLOS ONE’s publication criteria as it currently stands. Therefore, we invite you to submit a revised version of the manuscript that addresses the points raised during the review process.

We look forward to receiving your revised manuscript.

Kind regards,

Corstiaan den Uil

Academic Editor

PLOS ONE

"None"

6. Please amend the manuscript submission data (via Edit Submission) to include authors Catherine G. Derington, PharmD, MS; Jordana B. Cohen, MD, MSCE; April F. Mohanty, MPH, PhD; Tom H. Greene, PhD; James Cook, MS; Jian Ying, PhD; Guo Wei, MS; Jennifer S. Herrick, MS; Vanessa W. Stevens, PhD; Barbara E. Jones, MD, MS; Libo Wang, MD; Alexander R. Zheutlin, MD, MS; Andrew M. South, MD, MS; Thomas C. Hanff, MD, MSCE; Steven M. Smith, PharmD, MPH; Rhonda M. Cooper-DeHoff, PharmD, MS; Jordan B. King, PharmD, MS; G. Caleb Alexander, MD;l Dan R. Berlowitz, MD, MPH; Faraz S. Ahmad, MD, MS; M. Jason Penrod, MD; Rachel Hess, MD, MS; Molly B. Conroy, MD, MPH; James C. Fang, MD; Michael A. Rubin, MD, PhD, MS; Srinivasan Beddhu, MD; Alfred K. Cheung, MD; Weiming Xia, PhD; Lewis Kazis, ScD; William S. Weintraub, MD

Reviewers' comments:

Reviewer's Responses to Questions

**Comments to the Author**

1. Is the manuscript technically sound, and do the data support the conclusions?

Reviewer #1: Yes

Reviewer #2: Partly

2. Has the statistical analysis been performed appropriately and rigorously? 

Reviewer #1: Yes

Reviewer #2: Yes

3. Have the authors made all data underlying the findings in their manuscript fully available?

Reviewer #1: Yes

Reviewer #2: Yes

4. Is the manuscript presented in an intelligible fashion and written in standard English?

Reviewer #1: Yes

Reviewer #2: Yes

5. Review Comments to the Author

Reviewer #1: I commend you on this extensive work and nicely performed analysis. I personally agree with your conclusion that the decision on ACEi/ARB should be made very carefully. The risks vs benefits of RAAS should be weighed very carefully prior to considering in discontinuation. Yet, the controversy continues on this topic. I think it will be useful if the authors also cite the below paper:

Harky A, Chor CYT, Nixon H, Jeilani M. The controversy of using angiotensin-converting enzyme inhibitors and angiotensin receptor blockers in COVID-19 patients. J Renin Angiotensin Aldosterone Syst. 2021 Jan-Dec;22(1):1470320320987118. doi: 10.1177/1470320320987118. PMID: 33412991; PMCID: PMC7797594.

Reviewer #2: This retrospective study from the large Veterans Health Administration database compared outcomes in users of Angiotensin II type 1 receptor blockers (ARB) or Angiotensin converting enzyme inhibitors (ACEI) users (ARB/ACEI users) vs. non-ARB/ACEI users, and ARB users vs ACEI users, among SARS-CoV-2+ outpatients and COVID-19 hospitalized inpatients. Only hypertensive treated Veterans were included, and patients with compelling indications for ARBs or ACEIs were excluded. Adjusted comparisons between groups were done using propensity score-weighted Cox regression.

Among SARS-CoV-2+ outpatients, the rate of primary outcome (all-cause hospitalization or death) was lower in the ARB/ACEI users as compared to the non-users (adjusted HR 0.85, 95% CI 0.73-0.99), and it was also lower in ARB users vs ACEI users (HR 0.91, 95%CI 0.86-0.97).

Among COVID-19 hospitalized inpatients, the primary outcome (all-cause mortality) was not significantly different between groups.

The article is well written. The rationale for the objectives are accurate, and the hypothesis tested are important. The methods used are mostly appropriate and accurate, data were appropriately measured and analysed.

The results are only partly original. Many concordant studies have already showed that prognosis of patients hospitalized for COVID-19 is not affected by a prior exposure to ARB/ACEI, and the present study appears underpowered to be conclusive, due to the low number of inpatients ARB/ACEI users (n=210), and non-ARB/ACEI users (n=275).

The potential protective effect of a prior treatment with ARB/ACEI in SARS-CoV-2+ outpatients is a more original finding. However, several issues need to be addressed:

MAJOR

1) In the introduction, authors wrote: “However, most prior observational studies contained significant methodological limitations that reduce their overall validity, and thus, their clinical applicability and generalizability, including lack of a well-defined question, study population, or specific focus/exposure of interest which can lead to confounding and residual bias.”

This statement is severe for previous observational studies. I fear that the present study partly shares criticisms made of previous studies, particularly on the definition of populations, and the generalizability of the results and conclusions.

2) It is unclear why authors decided to exclude the "compelling indications" of ARB/ACEI for comparisons between ARB/ACEI users vs non users, and to include the compelling indications for comparisons between ARB vs ACEI.

This results in surprising numbers of patients included into analyses: the total number of patients treated with either an ARB or an ACEI (n=2,482 in the comparisons between ARB/ACEI users vs non users) appears 5 times lower than the sum of patients with an ARB and patients with an ACEI (n=13,581 in the comparisons between ARB vs ACEI). This may be confusing.

3) In the ARB/ACEI vs non ARB/ACEI comparison, the choice to exclude Veterans who had “compelling indications” for an ARB or an ACEI resulted in the exclusion of 78% of screened outpatients Veterans (18,067/23,036). What are the clinical applicability and generalizability of results observed in a highly selected population of hypertensive Veterans having no diabetes, chronic kidney disease, coronary heart disease, heart failure with reduced ejection fraction, or prior stroke (pages 7-8)?

4) Was multiple testing considered in choosing the significance levels? Is a P value of 0.035 (Table 1) significant considering multiple testing?

MINOR

1. How many of outpatients Veterans have been tested because of symptoms, and how many have been tested systematically without symptoms (screening, contact subject, travel)?

2. What were the criteria for hospitalization in the inpatients group? Did the criteria for hospitalization change from January to October 2020?

3. How many Veterans were common to the out- and in-patients cohorts?

6. PLOS authors have the option to publish the peer review history of their article (what does this mean?). If published, this will include your full peer review and any attached files.

Reviewer #1: No

Reviewer #2: No

---

## [Author Response · Author response to Decision Letter 0]

18 Feb 2021

REVIEWER #1

1. I commend you on this extensive work and nicely performed analysis. I personally agree with your conclusion that the decision on ACEi/ARB should be made very carefully. The risks vs benefits of RAAS should be weighed very carefully prior to considering in discontinuation. Yet, the controversy continues on this topic. I think it will be useful if the authors also cite the below paper: Harky A, Chor CYT, Nixon H, Jeilani M. The controversy of using angiotensin-converting enzyme inhibitors and angiotensin receptor blockers in COVID-19 patients. J Renin Angiotensin Aldosterone Syst. 2021 Jan-Dec;22(1):1470320320987118. doi: 10.1177/1470320320987118. PMID: 33412991; PMCID: PMC7797594.

Response: Thank you for your review and comment. We now cite the paper you mention both in the Introduction section and the Discussion section on pages 14 and 15, respectively. We also added citations for the recent published reports of REPLACE-COVID and BRACE CORONA randomized trials in the context of how our study adds value to the trial results that are now available. 

Introduction section page 5:

“During the ongoing pandemic, there has been unprecedented interest in whether ARBs and ACEIs are beneficial or harmful in patients with SARS-CoV-2 infection or coronavirus disease 2019 (COVID-19) [1].”

Discussion section page 15:

“Understanding the extent to which use of ARBs and ACEIs is beneficial or harmful in COVID-19 is a complicated yet critical knowledge gap because hypertension is highly prevalent among Americans with more severe COVID-19-related outcomes [1].”

Discussion section page 14:

“Two small, randomized trials reported no increased risk of harm with continuation of ARB or ACEI therapy among hemodynamically stable adults hospitalized for COVID-19 [2, 3]. The current report adds to these smaller trials by reinforcing the randomized trial results in a larger scale observational study in a diverse population. Also, our analysis evaluated SARS-CoV-2+ outpatients’ risk of developing COVID-19, a population not tested in these trials. Furthermore, our inpatient cohort was older, and our analysis directly compared ARB vs ACEI users, which these smaller trials were not adequately powered to evaluate. High-quality observational studies have potential to augment trial results and provide helpful insights into the association of ARB and ACEI use and COVID-19 outcomes.”

REVIEWER #2

1. This retrospective study from the large Veterans Health Administration database compared outcomes in users of Angiotensin II type 1 receptor blockers (ARB) or Angiotensin converting enzyme inhibitors (ACEI) users (ARB/ACEI users) vs. non-ARB/ACEI users, and ARB users vs ACEI users, among SARS-CoV-2+ outpatients and COVID-19 hospitalized inpatients. Only hypertensive treated Veterans were included, and patients with compelling indications for ARBs or ACEIs were excluded. Adjusted comparisons between groups were done using propensity score-weighted Cox regression. Among SARS-CoV-2+ outpatients, the rate of primary outcome (all-cause hospitalization or death) was lower in the ARB/ACEI users as compared to the non-users (adjusted HR 0.85, 95% CI 0.73-0.99), and it was also lower in ARB users vs ACEI users (HR 0.91, 95%CI 0.86-0.97). Among COVID-19 hospitalized inpatients, the primary outcome (all-cause mortality) was not significantly different between groups. The article is well written. The rationale for the objectives are accurate, and the hypothesis tested are important. The methods used are mostly appropriate and accurate, data were appropriately measured and analysed. The results are only partly original. Many concordant studies have already showed that prognosis of patients hospitalized for COVID-19 is not affected by a prior exposure to ARB/ACEI, and the present study appears underpowered to be conclusive, due to the low number of inpatients ARB/ACEI users (n=210), and non-ARB/ACEI users (n=275). The potential protective effect of a prior treatment with ARB/ACEI in SARS-CoV-2+ outpatients is a more original finding. However, several issues need to be addressed:

In the introduction, authors wrote: “However, most prior observational studies contained significant methodological limitations that reduce their overall validity, and thus, their clinical applicability and generalizability, including lack of a well-defined question, study population, or specific focus/exposure of interest which can lead to confounding and residual bias.” This statement is severe for previous observational studies. I fear that the present study partly shares criticisms made of previous studies, particularly on the definition of populations, and the generalizability of the results and conclusions.

Response: We understand the point the reviewer is making and agree that sentence should be modified. We have tempered the statement by modifying the first sentence and deleting the second sentence entirely. 

The sentence now reads as follows on page 5:

“However, several of the prior observational studies contain methodological limitations that may reduce their internal validity, and thus, their clinical applicability.” 

We provide specific examples of the methodological limitations we refer to later in the same paragraph.

2. It is unclear why authors decided to exclude the "compelling indications" of ARB/ACEI for comparisons between ARB/ACEI users vs non users, and to include the compelling indications for comparisons between ARB vs ACEI. This results in surprising numbers of patients included into analyses: the total number of patients treated with either an ARB or an ACEI (n=2,482 in the comparisons between ARB/ACEI users vs non users) appears 5 times lower than the sum of patients with an ARB and patients with an ACEI (n=13,581 in the comparisons between ARB vs ACEI). This may be confusing.

Response: We understand and agree that excluding those with compelling indications for the comparison of ARB/ACEI- vs. non-ARB/ACEI-based regimen users but not for the direct comparisons of ARB vs. ACEIs users led to significantly different samples sizes which may be confusing. However, we felt excluding those with compelling indications was critical for the comparison of ARB/ACEI- vs. non-ARB/ACEI-based regimen users and unnecessary for the direct ARB vs. ACEI comparison for the following reasons. We chose, a priori, to restrict the study population for the comparison of current users of ARB/ACEI- vs. non-ARB/ACEI-based regimen to only those without compelling indications because we believe that the confounding by indication for this specific comparison (ARB/ACEI users compared to non-users) would be insurmountable. We felt that patients with hypertension treated with ACEIs or ARBs are systematically different than those who are not on ACEIs or ARBs and that no level of statistical adjustment could overcome this systematic bias by indication. Instead, we chose to account for this bias through study design by restricting the analysis to only those without a history of compelling indications for ACEIs or ARBs. We felt that the restriction approach was a superior strategy to adjusting for the individual compelling indications as covariates in the full population because the latter approach cannot account for unmeasured confounders that likely coexist with the compelling indications that would go unaccounted for with statistical adjustment alone. We understand that although this approach, we believe, increases internal validity, it comes at the cost of external validity and generalizability to those with a history of compelling indications, which represents a large group of patients. 

We added the following sentence to the Limitations section of the manuscript on page 16 to address this limitation:

“Because we restricted the analysis comparing ARB/ACEI- vs. non-ARB/ACEI-based regimen users to only those without a compelling indication for an ARB or an ACEI to increase internal validity, our results may not be generalizable to adults with diabetes, chronic kidney disease, coronary heart disease, heart failure with reduced ejection fraction, or prior stroke.”

We felt that this restriction was not needed for the comparison of ARB vs. ACEI users because these compelling indications are shared between both medications; ACEIs and ARBs are used interchangeably as they are thought to be equivalent in benefit and safety [4, 5]. Therefore, we believe there is no need to restrict the population as the distribution of these conditions and the associated unmeasured factors are expected to be similar among ACEI and ARB users. 

We added the following sentences to the Methods section to state our rationale more clearly on page 7:

“For this comparison, because diagnoses other than hypertension (i.e., compelling indications) for ARB or ACEI use would introduce confounding by indication, we excluded Veterans with compelling indications defined in the 2017 American College of Cardiology/American Heart Association blood pressure guidelines [5] (i.e., diabetes, chronic kidney disease, coronary heart disease, heart failure with reduced ejection fraction, or prior stroke). We chose, a priori, to restrict the study population for the comparison of current users of ARB/ACEI- vs. non-ARB/ACEI-based regimen to only those without compelling indications because we believe that the confounding by indication for this specific comparison (ARB/ACEI users compared to non-users) would be insurmountable.

“Our second exposure compared current users of an ARB vs. ACEI (current user of an ARB vs. an ACEI). For this comparison, we included Veterans with and without compelling indications for ARBs and ACEIs, and we excluded Veterans who were current users of both an ARB and ACEI. For this comparison, restriction to those without compelling indications is not needed because these compelling indications are shared between both medications; ARBs and ACEIs are used interchangeably as they are thought to be equivalent in benefit and safety [4, 5]. As such, for this comparison, we included Veterans with and without compelling indications for ARBs and ACEIs, and we excluded Veterans who were current users of both an ARB and ACEI.”

3. In the ARB/ACEI vs non ARB/ACEI comparison, the choice to exclude Veterans who had “compelling indications” for an ARB or an ACEI resulted in the exclusion of 78% of screened outpatients Veterans (18,067/23,036). What are the clinical applicability and generalizability of results observed in a highly selected population of hypertensive Veterans having no diabetes, chronic kidney disease, coronary heart disease, heart failure with reduced ejection fraction, or prior stroke (pages 7-8)?

Response: Please see our response to Reviewer #2, Comment #2. We choose, a priori, to restrict the analysis comparing ARB/ACEI- vs. non-ARB/ACEI-based regimens users to only those without a compelling indication for an ARB or an ACEI to increase internal validity. By doing so, we understand this comes at the cost of external validity or generalizability of our results to groups with a history of compelling indications. 

We added the following sentence to the Limitations section to address this limitation on page 16:

“Because we restricted the analysis comparing ARB/ACEI- vs. non-ARB/ACEI-based regimen users to only those without a compelling indication for an ARB or an ACEI to increase internal validity, our result may not be generalizable to adults with diabetes, chronic kidney disease, coronary heart disease, heart failure with reduced ejection fraction, or prior stroke.”

4. Was multiple testing considered in choosing the significance levels? Is a P value of 0.035 (Table 1) significant considering multiple testing?

Response: We agree with the reviewer that multiple testing may increase the likelihood of false-positive findings. We did consider multiple testing in choosing the significance levels. In the current analysis, we have two main exposures assessed in two cohorts (i.e., the inpatient and outpatient cohorts). In each cohort for each exposure, we designate a single primary outcome and three secondary outcomes and multiple sensitivity analyses. In this case, we feel adjustment for multiple comparisons is not useful when presenting secondary outcomes or sensitivity analyses surrounding a single "main analysis." The secondary outcomes and sensitivity analyses are playing a supportive role to the main analysis, and generally provide the opportunity for greater caution in interpreting if the results are inconsistent between analyses, rather than an opportunity for declaring a result "significant" if any of the sensitivity analyses has a nominally significant p-value. As such, consistent with the recommendations from Rothman [6], Perneger [7], Feise [8], and Althouse [9], instead of performing an adjustment for multiple comparisons, we chose to provide extensive information and detail on the identification of the main analyses vs. sensitivity analyses in study design, methods, and statistical analyses and report effect sizes, confidence intervals, and p-values so that the readers can make their own assessments about the quality of our study and the interpretation of our findings in the context of the other recent reports on this topic. We posted our statistical analysis plan on clinicatrials.gov and we make note that additional dedicated studies are needed to confirm the results.

We added the following statement to the Limitations of the manuscript on beginning on page 16: 

“We did not adjust for multiple comparisons. Instead, consistent with the recommendations from Rothman [6], Perneger [7], Feise [8], and Althouse [9], we provided extensive information and detail on the study design, methods, and statistical analyses and report effect sizes, confidence intervals, and p-values so that the readers can make their own interpretation of our findings in the context of the other recent reports on this topic.”

5. How many of outpatients Veterans have been tested because of symptoms, and how many have been tested systematically without symptoms (screening, contact subject, travel)?

Response: Data on symptoms at the time of SARS-CoV-2 testing were unavailable. We added this to the limitations section on page 17:

“Data on symptoms at the time of SARS-CoV-2 testing were unavailable.”

6. What were the criteria for hospitalization in the inpatients group? Did the criteria for hospitalization change from January to October 2020?

Response: There were no standardized VA-system wide criteria for COVID-19 hospitalization during the pandemic. COVID-19 patient admission decisions likely varied by facility and over time according to clinical judgment, local bed availability, and patient acuity.

We added the following sentence to the Limitation section to address this on page 17:

 “Due to regional differences in the acceleration of COVID-19 cases and resource availability in the US, decisions on which patients with COVID-19 were admitted likely varied by facility and over time according to clinical judgment, local bed availability and patient acuity. Therefore, the severity of hospitalized patients with COVID-19 included in the inpatient analysis may have varied by region and over time.”

7. How many Veterans were common to the out- and in-patients cohorts?

Response: There were 452 Veterans who were included for analyses in both the outpatient and inpatient cohorts for the comparison of ARB/ACEI- vs. non-ARB/ACEI-based regimens users. For the comparison of ARB vs. ACEI users, there were 3,110 Veterans who were included for analyses in both the outpatient and inpatient cohorts.

Editorial Requests

1. We note that the grant information you provided in the ‘Funding Information’ and ‘Financial Disclosure’ sections do not match. When you resubmit, please ensure that you provide the correct grant numbers for the awards you received for your study in the ‘Funding Information’ section.

Response: We corrected the grant numbers. 

2. We note that you have indicated that data from this study are available upon request. PLOS only allows data to be available upon request if there are legal or ethical restrictions on sharing data publicly. For information on unacceptable data access restrictions, please see http://journals.plos.org/plosone/s/data-availability#loc-unacceptable-data-access-restrictions. If there are ethical or legal restrictions on sharing a de-identified data set, please explain them in detail (e.g., data contain potentially identifying or sensitive patient information) and who has imposed them (e.g., an ethics committee). Please also provide contact information for a data access committee, ethics committee, or other institutional body to which data requests may be sent.

Response: There are legal restrictions on sharing data publicly. The dataset used for this study contains sensitive patient information that is potentially identifying. The Department of Veterans Affairs (VA) restricts access to such data unless researchers meet specific criteria per VHA Directive 1605.01, Privacy and Release of Information, section 13e which includes a written request for the data, IRB approval with waiver for HIPAA authorization prior to the request for individually identifiable information, and approval by the VA. 

Response: We added Dr. Bress’s ORCID iD.

4. Please amend the manuscript submission data (via Edit Submission) to include authors Catherine G. Derington, PharmD, MS; Jordana B. Cohen, MD, MSCE; April F. Mohanty, MPH, PhD; Tom H. Greene, PhD; James Cook, MS; Jian Ying, PhD; Guo Wei, MS; Jennifer S. Herrick, MS; Vanessa W. Stevens, PhD; Barbara E. Jones, MD, MS; Libo Wang, MD; Alexander R. Zheutlin, MD, MS; Andrew M. South, MD, MS; Thomas C. Hanff, MD, MSCE; Steven M. Smith, PharmD, MPH; Rhonda M. Cooper-DeHoff, PharmD, MS; Jordan B. King, PharmD, MS; G. Caleb Alexander, MD;l Dan R. Berlowitz, MD, MPH; Faraz S. Ahmad, MD, MS; M. Jason Penrod, MD; Rachel Hess, MD, MS; Molly B. Conroy, MD, MPH; James C. Fang, MD; Michael A. Rubin, MD, PhD, MS; Srinivasan Beddhu, MD; Alfred K. Cheung, MD; Weiming Xia, PhD; Lewis Kazis, ScD; William S. Weintraub, MD

Response: We amended the manuscript submission via “Edit Submission” to include all co-authors.

Response: We included captions for our supporting information files at the end of our manuscript and updated in-text citations to match accordingly. 

 

REFERENCES

1. Harky A, Chor CYT, Nixon H, Jeilani M. The controversy of using angiotensin-converting enzyme inhibitors and angiotensin receptor blockers in COVID-19 patients. J Renin Angiotensin Aldosterone Syst. 2021;22(1):1470320320987118. Epub 2021/01/09. doi: 10.1177/1470320320987118. PubMed PMID: 33412991; PubMed Central PMCID: PMCPMC7797594.

2. Lopes RD, Macedo AVS, de Barros ESPGM, Moll-Bernardes RJ, Dos Santos TM, Mazza L, et al. Effect of Discontinuing vs Continuing Angiotensin-Converting Enzyme Inhibitors and Angiotensin II Receptor Blockers on Days Alive and Out of the Hospital in Patients Admitted With COVID-19: A Randomized Clinical Trial. JAMA. 2021;325(3):254-64. Epub 2021/01/20. doi: 10.1001/jama.2020.25864. PubMed PMID: 33464336; PubMed Central PMCID: PMCPMC7816106.

3. Cohen JB, Hanff TC, William P, Sweitzer N, Rosado-Santander NR, Medina C, et al. Continuation versus discontinuation of renin-angiotensin system inhibitors in patients admitted to hospital with COVID-19: a prospective, randomised, open-label trial. Lancet Respir Med. 2021. Epub 2021/01/11. doi: 10.1016/S2213-2600(20)30558-0. PubMed PMID: 33422263; PubMed Central PMCID: PMCPMC7832152.

4. Williams B, Mancia G, Spiering W, Agabiti Rosei E, Azizi M, Burnier M, et al. 2018 ESC/ESH Guidelines for the management of arterial hypertension. Eur Heart J. 2018;39(33):3021-104. Epub 2018/08/31. doi: 10.1093/eurheartj/ehy339. PubMed PMID: 30165516.

5. Whelton PK, Carey RM, Aronow WS, Casey DE, Jr., Collins KJ, Dennison Himmelfarb C, et al. 2017 ACC/AHA/AAPA/ABC/ACPM/AGS/APhA/ASH/ASPC/NMA/PCNA Guideline for the Prevention, Detection, Evaluation, and Management of High Blood Pressure in Adults: Executive Summary: A Report of the American College of Cardiology/American Heart Association Task Force on Clinical Practice Guidelines. Hypertension. 2018;71(6):1269-324. Epub 2017/11/15. doi: 10.1161/HYP.0000000000000066. PubMed PMID: 29133354.

6. Rothman KJ. No adjustments are needed for multiple comparisons. Epidemiology. 1990:43-6.

7. Perneger TV. What's wrong with Bonferroni adjustments. Bmj. 1998;316(7139):1236-8.

8. Feise RJ. Do multiple outcome measures require p-value adjustment? BMC medical research methodology. 2002;2(1):1-4.

9. Althouse AD. Adjust for Multiple Comparisons? It's Not That Simple. Ann Thorac Surg. 2016;101(5):1644-5. Epub 2016/04/24. doi: 10.1016/j.athoracsur.2015.11.024. PubMed PMID: 27106412.

---

## [Editor Report · Decision Letter 1]

19 Feb 2021

Angiotensin II receptor blocker or angiotensin-converting enzyme inhibitor use and COVID-19-related outcomes among US Veterans

PONE-D-21-01248R1

Dear Dr. Bress,

We’re pleased to inform you that your manuscript has been judged scientifically suitable for publication and will be formally accepted for publication once it meets all outstanding technical requirements.

Kind regards,

Corstiaan den Uil

Academic Editor

PLOS ONE
---

## [Editor Report · Acceptance letter]

13 Apr 2021

PONE-D-21-01248R1 

Angiotensin II receptor blocker or angiotensin-converting enzyme inhibitor use and COVID-19-related outcomes among US Veterans 

Dear Dr. Derington:

I'm pleased to inform you that your manuscript has been deemed suitable for publication in PLOS ONE. Congratulations! Your manuscript is now with our production department. 

Kind regards, 

on behalf of

Dr. Corstiaan den Uil 

Academic Editor

PLOS ONE